# Segmentation of dense and multi-species bacterial colonies using models trained on synthetic microscopy images

**Vincent Hickl** [iD][1,2,3]*, **Abid Khan**[4,5], **René M. Rossi** [iD][3], **Bruno F. B. Silva**[1,2,3], **Katharina Maniura-Weber**[1]

**1** Laboratory for Biointerfaces, Empa, Swiss Federal Laboratories for Materials Science and Technology, St. Gallen, Switzerland, **2** Center for X-ray Analytics, Empa, Swiss Federal Laboratories for Materials Science and Technology, St. Gallen, Switzerland, **3** Laboratory for Biomimetic Membranes and Textiles, Empa, Swiss Federal Laboratories for Materials Science and Technology, St. Gallen, Switzerland, **4** Department of Physics, University of Illinois Urbana-Champaign, Urbana, Illinois, United States of America, **5** NASA Ames Research Center, Moffett Field, California, United States of America

* vincent.hickl@empa.ch

**Data availability statement:** All real and raw synthetic images used in the figures above and

## Abstract

The spread of microbial infections is governed by the self-organization of bacteria on surfaces. Bacterial interactions in clinically relevant settings remain challenging to quantify, especially in systems with multiple species or varied material properties. Quantitative image analysis methods based on machine learning show promise to overcome this challenge and support the development of novel antimicrobial treatments, but are limited by a lack of high-quality training data. Here, novel experimental and image analysis techniques for high-fidelity single-cell segmentation of bacterial colonies are developed. Machine learning-based segmentation models are trained solely using synthetic microscopy images that are processed to look realistic using a state-of-the-art image-to-image translation method (cycleGAN), requiring no biophysical modeling. Accurate single-cell segmentation is achieved for densely packed single-species colonies and multi-species colonies of common pathogenic bacteria, even under suboptimal imaging conditions and for both brightfield and confocal laser scanning microscopy. The resulting data provide quantitative insights into the self-organization of bacteria on soft surfaces. Thanks to their high adaptability and relatively simple implementation, these methods promise to greatly facilitate quantitative descriptions of bacterial infections in varied environments, and may be used for the development of rapid diagnostic tools in clinical settings.

## Author summary

Bacteria organize themselves on surfaces in ways that influence the spread of infections, but studying these behaviors is difficult, especially when multiple species are involved

to train CycleGAN are freely available on Zenodo: https://doi.org/10.5281/zenodo.12759487. The Python code to produce raw synthetic images and to reproduce the analysis in this article is available on GitHub: github.com/vhickl/synth-bacteria-segmentation. All code required to train CycleGANs to process synthetic images, to calculate FID scores, and generate processed synthetic images using trained models is available on GitHub: github.com/abid1214/cyclegan.

**Funding:** The author(s) received no specific funding for this work.

**Competing interests:** The authors have declared that no competing interests exist.

or their environment is complex. To address this challenge, we develop a new approach using machine learning to analyze bacterial colonies at the single-cell level. Our method uses artificial, computer-generated images to train the system, eliminating the need for complicated biophysical models. This approach provides new insights into the spatial organization of bacterial colonies, even when they are densely packed and the imaging conditions are less than ideal. We apply this technique to images containing multiple species of bacteria and find that it is very effective for both brightfield and confocal microscopy. The insights gained from this work could help scientists better understand how bacterial infections develop and spread, and may lead to faster diagnostic tools for clinical use. Our method is versatile and easy to implement, making it suitable for a wide range of imaging applications.

## Introduction

Microbial attachment and aggregation at surfaces are fundamental to the resilience of bacterial infections. After planktonic bacteria settle onto a substrate, they form microcolonies of small numbers of cells that become precursors for the biofilms [1,2] that make infections dramatically more resistant to medical intervention. The self-organization of these bacteria during the early stages of infection plays a crucial role in determining the progression of the infection [3–7]. Understanding how bacterial attachment and microcolony formation vary across different surfaces reveals strategies to prevent infections before they fully develop [8–10]. In turn, understanding the mechanics of biofilm architecture at the single-cell scale is essential to understanding how to disrupt their proliferation and create novel therapies that penetrate the bacteria's natural defenses.

Novel imaging and analysis methods are needed to study bacterial self-organization in clinically relevant systems. The importance of single-cell segmentation for understanding the development of bacterial colonies is well understood, and some recent studies have achieved accurate segmentation of dense bacterial colonies and biofilms [11–14]. However, to obtain optimal imaging conditions, the colonies in these studies are grown directly on glass coverslips, which does not correspond to the far more complex environments bacteria encounter in clinical settings. Developing methods for image analysis at surfaces with different geometric, material, and chemical properties has been a crucial challenge [15,16].

In principle, state-of-the-art segmentation methods using machine learning can overcome this problem, but typically require tedious human annotation of large image sets for training (meaning many cell labels must be drawn by hand in each image). Creating accurate segmentation models requires training data that closely resembles the data to be analyzed, dramatically increasing the amount of annotated images needed to train models for different applications [17]. As this problem transcends the study of bacteria and is relevant to a number of different disciplines, some current research in the field of image segmentation is focused on producing training datasets without human annotation. One promising approach is the use of artificial images that resemble experimental images closely enough to train segmentation models [18–20]. While some studies have applied this approach to bacterial colonies and biofilms, they require complex, explicit computational models of cell growth and of the imaging apparatus, experimentally measured point-spread functions, or extensive post-processing, making them challenging to apply to a broad range of experimental conditions [21–24]. Simple, adaptable, and accessible computational tools are sorely needed to empower researchers to develop segmentation models tailored to their specific experimental system.

Existing studies of bacterial self-organization and novel methods for single-cell segmentation have mostly dealt with colonies consisting of a single species [7,12,21,25]. However, the vast majority of real infections involve multiple species that collaborate or compete in complex ways [26,27], and have profound effects on the efficacy of different treatments [19,28–30]. While species can be distinguished by using mutant strains expressing different fluorescent proteins, or through the use of different fluorescent stains, these methods greatly complicate experimental protocols and limit their relevance to clinical settings. The mechanical interactions between multiple bacterial species at the microscale remain poorly understood, partially because there are few available analysis methods that can perform single-cell segmentation and distinguish between strains based on morphology alone [31–33]. It is thus crucial that novel image segmentation methods be developed to include the capacity for multi-species segmentation.

Here, we present a new method for creating single-cell segmentation models from synthetic microscopy images produced through image-to-image translation. Using a custom microfluidic device, dense monolayers of rod-shaped bacteria are grown on PDMS films. In a different set of experiments, mixed suspensions of rod-shaped and spherical bacteria were imaged. Synthetic microscopy images of densely packed and multi-species bacterial colonies are produced using a simple and adaptable model. These 'raw' synthetic images are then processed using a cycle generative adversarial network (cycleGAN) [34,35] to resemble real images and serve as a training set for custom segmentation models. Thus, bespoke segmentation models adapted to a variety of experimental conditions can be trained quickly without human annotation. Dense monolayers of rod-shaped cells grown on soft substrates not optimized for high signal-to-noise ratios can be segmented with greater accuracy than with existing models from the literature. Quantitative information on the distribution of the bacteria can then be used to gain novel insights into bacterial self-organization. In images of mixed colonies, cells of different species can be automatically identified in datasets from both confocal and brightfield microscopy. This approach to analyzing bacterial colonies promises to greatly simplify the creation of accurate segmentation models tailored to a variety of in-vitro and in-vivo systems.

## Materials and methods

### Bacteria cultures on PDMS in microfluidic device

Single-species cultures of *Pseudomonas aeruginosa* (*P. a.*) were grown on top of thin polydimethylsiloxane (PDMS) sheets in a microfluidic device. *P. a.* MPAO1 *flgE* knockout mutants, constructed as previously described [36], were grown overnight in 30% tryptic soy broth (TSB) with 0.25% glucose. The culture was supplemented with fresh medium to dilute it to $OD$ = 0.2, and then further diluted 50× in medium, resulting in identical starting concentrations across experiments.

To construct a flow chamber, thin PDMS sheets were formed by pouring 0.55 g of Sylgard 184 at a base to curing agent ratio of 10:1 into a 8.5 cm diameter petri dish. After degassing in a vacuum chamber, the sheets were cured for 2 hrs at 80°C, resulting in sheets of thickness approximately 100 μm. A 2 × 1 cm piece of the resulting film was then cut out and placed on a #1.5 coverslip. On either side of this PDMS film, two pieces of 1 mm thick PDMS (formed using the same procedure as the thin films) were placed on the coverslip as spacers, and polyethylene tubes with an outer diameter of 1 mm (Huberlab), were attached to the coverslip on either side of the PDMS film. Then, 0.5 mL of the diluted *P.a.* was pipetted onto the thin PDMS, seeding the surface with some attached cells for subsequent growth. Finally, another

coverslip was placed on top of the spacers, and the edges were sealed with epoxy, forming a sealed microfluidic flow chamber for controlled growth of bacteria on the PDMS film.

Bacteria were grown on the PDMS overnight under constant flow (0.01 mL/min) from a syringe pump of fresh growth medium supplemented with 1 μM SYTO 9 nucleic acid stain. The steady addition of SYTO 9 at a low concentration ensured that all bacteria were stained while preserving their viability. The resulting dense monolayers of *P.a.* were then imaged using a Zeiss LSM780 confocal microscope with a 63× oil immersion objective.

### Multi-species staining and imaging

To test the simultaneous segmentation of multiple species, *P. a.* and *Staphylococcus aureus* (*S. a.*) strain ATCC 6538 were cultured overnight as described above. The cultures were separately centrifuged at 7000g for 10 minutes, and the pellets were resuspended in phosphate-buffered saline (PBS). For confocal imaging, some samples were stained with 2.5 μM SYTO 9. Equal volumes of both suspensions were mixed together via vortexing, and 5 μL of the mixture was placed between two coverslips for imaging. Confocal imaging was performed as described above, and brightfield imaging was performed using a Nikon Eclipse Ti2 microscope at 40× magnification.

### Synthetic images for segmentation model training

Raw synthetic images of bacteria at interfaces were created using custom programs written in Python, as described in Section 1 in S1 Appendix. In brief, to model rod-shaped *P. a.* cells, bright rectangles with circular caps were drawn with various positions and orientations on a dark background. In some images, cells were drawn with random orientations, while in others, they were aligned parallel to their nearest neighbors to simulate the tendency of bacteria to exhibit orientational order. The degree of alignment was varied by introducing a random noise term to the angle of each cell. Images with varying degrees of alignment were produced in each dataset to improve realism and ensure the training data for the segmentation model were not biased towards highly aligned or randomly aligned cells, which is important for the analysis of bacterial self-organization below. *S. a.* cells were modeled as circular disks. In some images, cells were drawn in clusters to include regions of more densely packed and more dilute cells. Additionally, a maximum overlap parameter was implemented to ensure neighboring cells could touch but did not overlap excessively. Cell dimensions were chosen to match those in a randomly chosen, hand-measured sample of cells from real experimental images. Similarly, the cell density in raw synthetic images was chosen to roughly match the density observed in experimental images.

Simultaneously, this program also produced a label mask for each synthetic image – an integer array of the same dimensions as the image in which the set of pixels corresponding to each cell is assigned a unique integer value. These masks were later used to train the segmentation model. Initially, raw synthetic images were simply binary masks – white cells on a black background. Gaussian noise (standard deviation = 0.1) was added to these images to ensure enough variability across the synthetic data set. The addition of noise was performed within the program used to the train cycleGANs, described below. Throughout this paper, the terms 'raw synthetic image' and 'processed synthetic image' are used, respectively, to refer to these images before and after they are transformed by cycleGAN to resemble real images.

## Processing synthetic images using cycleGAN

A cycleGAN was trained to use one of its components: the generator that inputs a raw synthetic image and outputs a processed synthetic image. Training a cycleGAN requires two unpaired datasets $X$ and $Y$ between which images are to be translated. The network consists of four models: two discriminators and two generators. The generators $G_Y : X \rightarrow Y$ and $G_X : Y \rightarrow X$ take an image $x, y$ from one domain and process it to resemble an image belonging to the other domain $G_Y(x), G_X(y)$. The discriminators $D_X : X \rightarrow \mathbb{R}, D_Y : Y \rightarrow \mathbb{R}$ to take an image $\hat{x}, \hat{y}$ and output whether it belongs in their respective domains $D_X(\hat{x}), D_Y(\hat{y})$. The two generators and two discriminators train by competing with each other: the generators are trained to fool the discriminators, and the discriminators are trained to tell apart generator images from real images.

The model architectures and losses used in this work were the same that were used in [18]. The training set of cycleGAN is composed of experimental and synthetic bacterial images. To prepare the cycleGAN training set, all the image intensities were normalized to [-1, 1] within their group for each data set. The images were then cut into $256 \times 256$ patches. The batch size was 42, however, when adding each patch into the batch, they were randomly augmented by a combination of rotation (multiples of 90 degrees) and flipping (vertically or horizontally). All networks were trained for 400 epochs, where in the first 200 epochs, the learning rate of the generators was $2 \times 10^{-5}$, and in the last 200 epochs, the learning rate linearly decayed to zero. In the training procedure, it is important for the discriminators not to excessively outperform the generators so that the generators can train better against the discriminators. To ensure this, the learning rate of the discriminators $\eta_{\text{disc}}$ was set to be varied at each training step,

$$\eta_{\text{disc}} = 2\eta_{\text{gen}} |1 - \alpha_{\text{disc}}|$$

where $\eta_{\text{gen}}$ is the learning rate of the generators, and $\alpha_{\text{disc}}$ is the accuracy, or fraction of the discriminators correctly evaluating test images.

It is not always true that the trained model at the last epoch provides generated images that most closely resemble the real image dataset. To obtain the best model in our training process, a dataset of processed synthetic images was constructed every 10 epochs. Then, the Fréchet inception distance (FID) between the real dataset and each processed synthetic dataset is calculated [37]. The model whose processed image dataset gives the lowest FID relative to the real dataset was then chosen.

## Segmentation model training

Segmentation models were trained using the synthetic microscopy images generated by the cycleGAN using the Omnipose package for Python [11]. This package is open-source and freely available (github.com/kevinjohncutler/omnipose, installed development version April 30th, 2024). Details and training parameters can be found in the SI. The model trained on real, hand-annotated images of bacteria against which the performance of our algorithm was compared ('Bact_fluor_omni') is included when installing Omnipose.

For the model trained to segment monolayers of *Pseudomonas aeruginosa* (*P. a.*), the training data consisted of 126 processed synthetic images of dimensions $512 \times 512$ pixels and their corresponding label masks. For the models trained to segment cells in mixed colonies of *P. a.* and *Staphylococcus aureus* (*S. a.*) imaged using confocal microscopy, the training data consisted of 226 processed synthetic images of dimensions $256 \times 256$ pixels. For the

models trained to segment cells in mixed colonies imaged using brightfield microscopy, the training data consisted of 441 processed synthetic images of dimensions $256 \times 256$ pixels.

For segmentation of multi-species colonies, the synthetic images used were the same for the models trained to detect rods and circles, but in each case the corresponding masks only contained cells of a single species (rods or circles, respectively). The two models trained to segment *P. a.* and *S. a.* cells, respectively, were then combined in single Python program to perform simultaneous segmentation and classification. In most test images used here, a small number of cells was identified by both models. These cells were assigned to the species for which the false positive rate was lower overall in that set of test images. Here, for both confocal and brightfield microscopy, this means that cells identified by both models were classified as *S. a.* by our program.

## Calculating segmentation model performance

The performance of the segmentation models was evaluated by calculating several metrics that can then be used to compute a single parameter, "Panoptic quality" (PQ), which ranges from 0 to 1 and provides a general metric for the quality of the segmentation [38]. PQ is defined as

$$PQ = \frac{\sum_{(g,c) \in TP} IoU(g,c)}{|TP|} \times \frac{|TP|}{|TP| + \frac{1}{2}|FP| + \frac{1}{2}|FN|}, \tag{1}$$

where $g$ and $c$ are ground truth and candidate cells, respectively, $IoU$ is the intersection over union of each ground truth and candidate pair, and the sum is taken only over pairs for which $IoU > 0.5$, meaning the candidate is a true positive (TP). $|TP|$, $|FP|$, and $|FN|$ are the number of true positives (candidate cells from the segmentation model that correspond to actual cells), false positives (candidate cells that do not correspond to real cells), and false negatives (real cells not identified by the model). The first term in the $PQ$ is sometimes called "segmentation quality" and measures how well true positives match with the corresponding real cell at the pixel level, on average. The second term is the "recognition quality," which measures the model's ability to correctly find true positives. The recognition quality (RQ) is also known as the $F_1$ score. It is similar to the Jaccard index, another similarity coefficient in which false positives and negatives have a stronger effect. Thus, $PQ$ provides a comprehensive metric for evaluating the performance of a segmentation model.

## Quantifying cell alignment in *P. a.* monolayers

The results of the single-cell segmentation of densely-packed, rod-shaped *P. a.* were used to extract quantitative information about the bacteria's self-organization. All analyses were performed with custom Python code using open-source and freely available libraries. The `regionprops` function from the Sci-kit image analysis package was used to extract the positions, orientations (measured here so that the orientation angle $\theta = 0$ when a rod-shaped cell is aligned with the vertical axis), and dimensions of each cell. Cells in contact with the edge of the image or below a size threshold of 10 pixels were removed prior to further analysis. Additionally, when calculating local cell densities, only cells at least 5.74 μm (twice the average cell length) from the nearest edge of the image were included in the analysis.

## Results

### A novel approach to bacterial segmentation with synthetic microscopy images and cycleGAN

The purpose of the methodology presented here is to provide an efficient and adaptable way to create image segmentation models in the life sciences, with a particular focus on microscopy applications for the study of bacterial infections. Our approach is summarized in Fig 1. First, real microscopy images of bacteria are recorded with custom imaging setups and multiple microscopy techniques. Then, custom computational models are used to create 'raw' synthetic images of bacteria - images in which cell densities and shapes are approximately equal to those in the real images but that do not contain noise, blurred edges, anisotropic intensities, and other optical imperfections characteristic of real imaging techniques. A 'mask' is produced by the model alongside each raw synthetic image to encode the location, morphology, and species of each cell. The real and raw synthetic images are then used as inputs for a cycleGAN, used here to 'process' synthetic images by giving them optical characteristics to resemble the real images. Together with the original masks of the raw synthetic images,

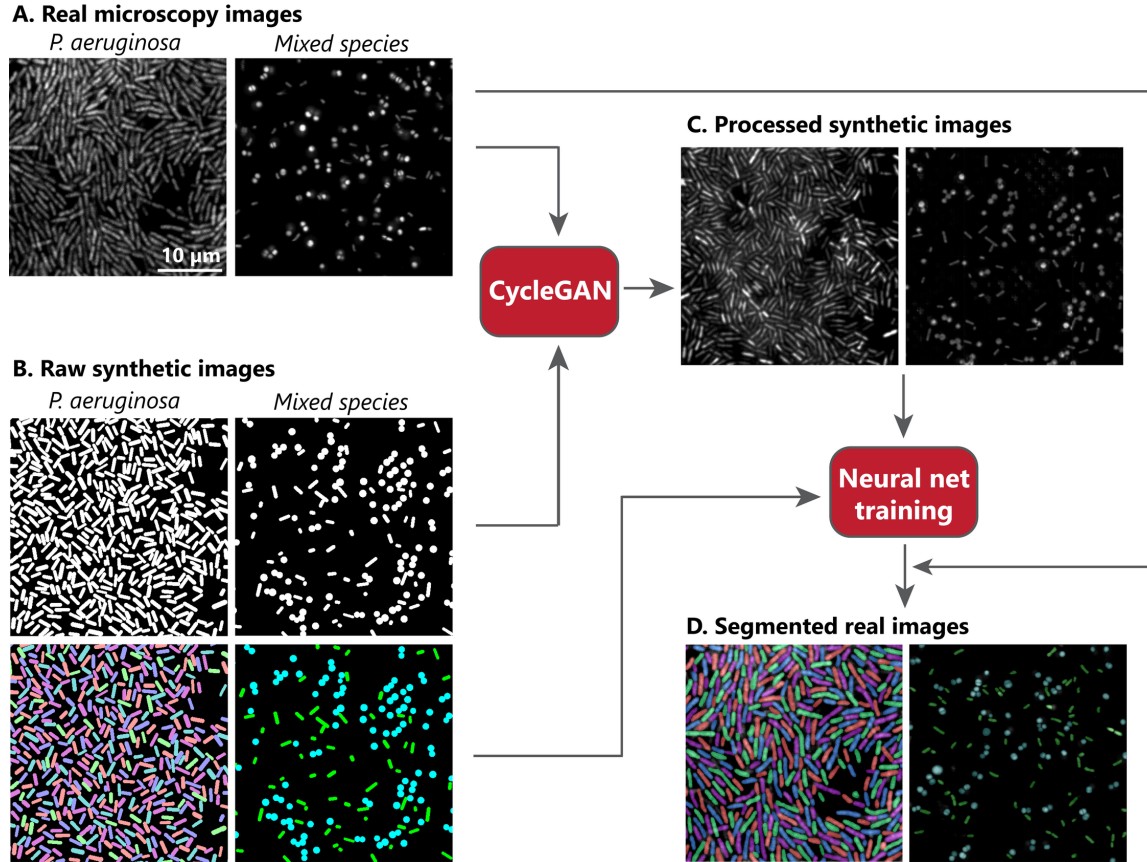

**Fig 1. Workflow of segmentation model creation and application.** (A) Real images of bacterial monolayers are taken using confocal microscopy. (B) Custom python algorithms are used to produce 'raw' synthetic images along with ground truth label masks. (C) Using cycleGAN, raw synthetic images are processed to qualitatively resemble real images. (D) The processed synthetic images and their masks are used to train a segmentation model in Omnipose [11]. This model is then used to segment real images. The scale bar is 10 μm, and the scale is the same for all images.

these processed synthetic images are then used to train neural networks to perform single-cell segmentation and species classification on real images.

## Imaging dense single-species and mixed colonies of bacteria at interfaces

To demonstrate the power and adaptability of our single-cell segmentation approach, experiments were conducted with colonies of clinically relevant bacterial strains using multiple sample preparation and imaging methods. Densely packed, single-species colonies of *Pseudomonas aeruginosa* (*P. a.*) were grown in a custom microfluidic device on 0.1 mm thin films of PDMS (Fig 2, top). PDMS was chosen because it is an ideal model substrate used to study attachment and mechanosensing of bacteria at surfaces with different mechanical and geometric properties [8,10,39]. Bacteria were left to grow overnight in the chamber under a constant flow of medium supplemented with nucleic acid stain SYTO 9 for fluorescent imaging. The flow served to provide ample nutrients for continued growth and to wash away any cells not attached to the PDMS. The resulting dense monolayers were imaged through the PDMS (Fig 2, top right). PDMS has a different refractive index than the glass and the immersion oil used, and attenuates the incident and emitted light to and from the sample. Impurities within the PDMS may also interfere with sample illumination. These factors all contribute to a reduction in the signal-to-noise ratio and an increase in the point spread function (PSF) of the imaging system. These suboptimal imaging conditions (compared to bacterial colonies imaged directly on a glass cover slip) were chosen deliberately to develop a segmentation method that can provide quantitative information for a variety of experimental protocols involving bacteria at different surfaces.

To develop a method for simultaneous segmentation and classification of multiple strains, multi-species bacterial colonies with undifferentiated staining were formed. Liquid cultures of *P. a.* and *Staphylococcus aureus* (*S. a.*) grown overnight were washed and resuspended in fresh PBS, mixed together, and either imaged immediately with brightfield microscopy or

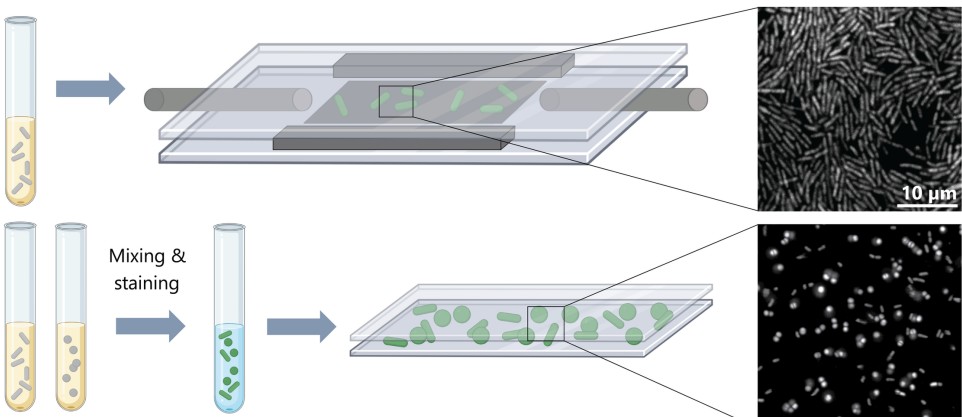

**Fig 2. Experimental methods.** Top: *Pseudomonas aeruginosa* (*P. a.*) are grown in liquid culture overnight, diluted in fresh medium, and seeded onto a thin PDMS film inside a custom-built microfluidic chamber. There, bacteria are grown overnight under a constant flow of medium with a low concentration of SYTO 9 to form a dense monolayer on the PDMS, which is then imaged using confocal microscopy. Scale bar is 10 μm. Bottom: *Staphylococcus aureus* (*S. a.*) and *P. a.* are grown separately overnight, washed, resuspended in fresh PBS, mixed, and stained with SYTO 9. A small volume of the suspension is then placed between two coverslips and imaged using confocal microscopy. Scale bar is 10 μm. Figure created in part using BioRender.

stained with SYTO 9 for confocal imaging (Fig 2, bottom). A droplet of this mixed suspension was then placed between two coverslips and imaged directly with a confocal microscope. Separately, unstained samples were similarly mounted for brightfield imaging. The resulting monolayers were less dense than the single-species monolayers of *P. a.*, although clusters of cells could still be found. Both rod-shaped cells (*P. a.*) and spheroidal cells (*S. a.*) can clearly be observed. These images are then used by an image-to-image translation algorithm to process synthetic images which in turn are used to train segmentation models.

## Rapid training of segmentation models using synthetic images and image-to-image translation

Synthetic microscopy images of both single-species and mixed bacterial colonies are created to provide training data for segmentation models without human annotation. Using custom python algorithms, rod-shaped bacteria (*P. a.*) or mixtures of rod-shaped and spherical bacteria (*S. a.*), are drawn as bright shapes on a dark background, forming 'raw' synthetic images (Fig 1B). The cell density, degree of alignment (for rod-shaped bacteria), and degree of clustering were varied to provide a diverse dataset that includes the various cell configurations observed in the experimental images.

Raw synthetic images are processed to qualitatively resemble the real images without altering the ground truth of cell positions and orientations. Real experimental and raw synthetic images, created as described above, are used together to train a cycleGAN, which produces a generator that transforms synthetic images to closely resemble real experimental ones (Fig 1C). Separate cycleGANs are trained for single-cell and mixed colonies. The raw synthetic images are modified to add noise, vary the brightness of the cells, and blur near cell boundaries or edges. However, the cells' positions and orientations are preserved, meaning the masks created for the raw synthetic images can later be used to train a segmentation model.

From the processed synthetic images and corresponding masks, tailored segmentation models are created to segment and classify bacteria in each set of real images. Models are trained using the general image segmentation tool Omnipose [11], which provides a framework to train deep neural network algorithms for bacterial segmentation. For our application, the training data consists of the synthetic images processed by cycleGAN along with their corresponding masks (Fig 1D). No real experimental images or other hand-annotated images were used for this training. Three different segmentation models were trained here: one for monolayers of *P. a.* grown on PDMS, a second to identify and segment *P. a.* in mixed colonies with *S. aureus*, and a third for *S. aureus* in the same mixed colonies. The latter two models, when used together, allow for simultaneous classification and segmentation of both species in mixed colonies without differential staining.

## Accurate segmentation of dense *P. aeruginosa* monolayers using synthetic images

Segmentation models trained on processed synthetic images of single-species colonies of *P.a.* produce excellent single-cell segmentation of densely-packed colonies. Several experimental images, including densely packed and more dilute monolayers of cells (Fig 3A), were manually labeled to test the accuracy of this segmentation model. These manual labels provided 'ground truth' masks to which masks from the segmentation model could be compared (Fig 3B). In total, 13 images containing a total of 1533 cells were manually annotated. In all test images, the masks generated by the model trained on processed synthetic images ('synthetic model') accurately reproduce the labels in the ground truth mask. Cells are accurately

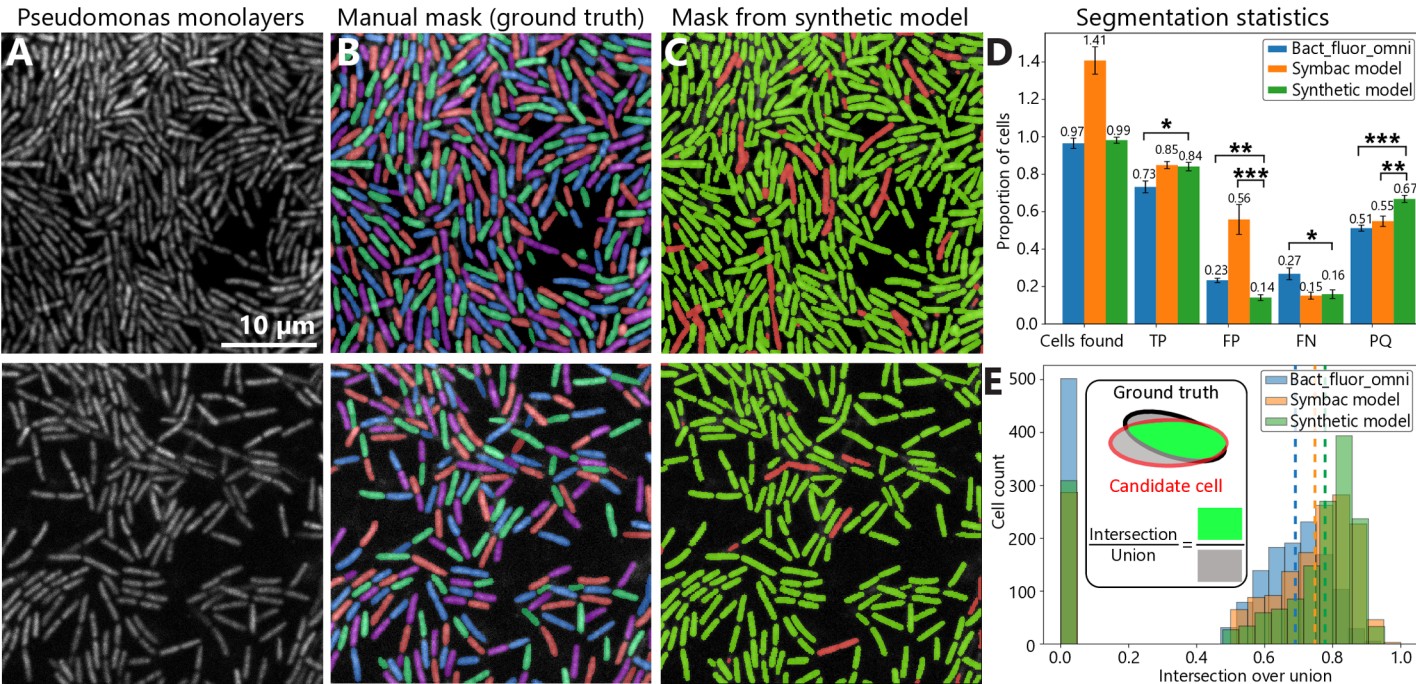

**Fig 3. Segmentation of *P. a.* monolayers.** (A) Confocal microscopy images at 63× magnification of dense (top) and dilute (bottom) monolayers of *P. a.* stained with SYTO 9. The scale bar is 10 μm, and the scale is the same for all images. (B) Sample hand-annotated masks showing the ground truth positions of all bacteria in the monolayers. Colors are only used to visually distinguish cells and do not correspond to any physical parameters. (C) Sample masks produced by the segmentation model trained on synthetic microscopy images processed with cycleGAN ('synthetic model'). Correctly and incorrectly segmented cells are colored in green and red, respectively. (D) Segmentation statistics and comparison to 'Bact_fluor_omni', a model trained on high-quality hand-annotated experimental images of bacteria [11], and a model trained on synthetic images created using SyMBac [24]. True positives (TP), false positives (FP), and false negatives (FN) are given as a proportion of the number of cells in the ground truth mask, and panoptic quality (PQ) [38] is defined between 0 and 1. Error bars represent standard error from variation between images. The data are drawn from 13 different images such as the ones in panels (B) and (C), and include 1533 cells in total. Asterisks represent significance from a t-test at the *P* < 0.05, 0.01, and 0.001 levels, respectively. (E) Distribution of intersection over union (which measures how accurately cells are identified by comparing candidate cells in the segmentation mask to the ground truth at the pixel level) from our model trained synthetic images by cycleGAN, Bact_fluor_omni, and the SyMBac model. The bars at *IoU* = 0 represent the number of false negatives. Dashed lines represent average IoUs. Inset: diagram of intersection over union for a model cell.

and reliably distinguished from the background, and nearby cells are distinguished from each other with few exceptions, while individual cells are rarely incorrectly divided into smaller ones (Fig 3C).

Several quantitative metrics are calculated to assess the quality of the segmentation model. These include the number of total cells found by the segmentation model ('candidate cells') as well as the number of true positives (TP), false positives (FP), and false negatives (FP), all as a proportion of cells in the ground truth mask (Fig 3D). The 'panoptic quality' (PQ) is also computed, which is a common measure of overall segmentation quality that takes into account the proportion of cells correctly identified and how well the candidate cells match the corresponding cell in the ground truth [38]. The number of cells identified by the synthetic model is, on average, 99 ± 3% of the number of cells in the ground truth mask, suggesting neither under- nor oversegmentation. On average, 84% of cells are correctly identified (true positives). Additionally, the intersection over union (IoU) is computed for each cell, which measures how well the candidate cell matches the ground truth (Fig 3E, inset). True positives are defined as candidate cells with *IoU* > 0.5. This IoU threshold guarantees that at most one

true positive exists for each ground truth cell. The accuracy with which cell positions, orientations, aspect ratios, and areas can be measured was analyzed as a function of IoU (see Section 4 in S1 Appendix). It was found that these important characteristics can be reliably measured using a threshold of $IoU > 0.5$, as shown in Fig E in S1 Appendix.

The segmentation model trained on processed synthetic images outperforms existing state-of-the-art segmentation models available in the recent literature. Here, the model trained on synthetic images processed by cycleGAN ('synthetic model') is compared to 'Bact_fluor_omni', a model trained in Omnipose using hand-annotated experimental images of various bacteria strains with different densities and morphologies, as shown in Fig 3D and 3E. This model is optimized for high-quality fluorescent images and similar models have been shown to outperform other recent segmentation techniques [11]. Additionally, the model presented here is compared to a 'retrained' model trained on synthetic images created by SyMBac [23,24]. These synthetic images are created using sophisticated modeling of fluorescent emitters inside each bacterium, and further processed using experimentally measured or theoretically computed PSFs. Across all calculated metrics, our model trained on processed synthetic images performs significantly better than Bact_fluor_omni. On average, the percentage of cells correctly identified is 11 percentage points higher, the panoptic quality is 16 percentage points higher, and the mean IoU of candidate cells from the synthetic model is 12% higher compared to those from the model trained on a high quality set of hand-annotated experimental images, on average. Additionally, cell areas and aspect ratios can be measured more accurately using the model trained on processed synthetic images, even when the IoU is the same for both models, as shown in Fig E in S1 Appendix.

For the test images obtained here, the synthetic model trained on images processed by cycleGAN outperforms the model trained on images from SyMBac. These two models produce equal proportions of true positives and false negatives. However, the SyMBac model produces a large number of false positives, suggesting significant oversegmentation of cells. As a result, the PQ is significantly better for the model trained on images processed by cycleGAN. The mean IoU of candidate cells from the 'synthetic model' is 3% higher than that of the SyMBac model, on average. When controlling for IoU, measurements of cell positions, orientations, and aspect ratios are comparable across these two models trained on different synthetic images, but measurements of cell areas are more accurate using the model trained on images processed by cycleGAN, as shown in Fig E in S1 Appendix. It should be noted that the segmentation model trained on images from SyMBac is intended to be a bespoke model for a particular set of experimental and imaging conditions, which may differ from the ones described here. The PQs of all three models decrease when the IoU cutoff is increased. For all IoU cutoffs, the model trained on processed synthetic images performs better than the two models from the literature.

## Quantifying bacterial self-organization in dense *P. aeruginosa* monolayers

Accurate single-cell segmentation enables quantitative statistics about bacterial self-organization at the cellular scale to be determined. Using the masks provides by the segmentation model trained on processed synthetic images, the positions, orientations, and dimensions of all cells in densely packed monolayers of *P. a.* are determined. Since the bacteria in this experiment have no front or back, they have 180° symmetry, and their orientations are given by an angle $\theta$ in the interval $(-\pi/2, \pi/2]$ (Fig 4A). The parallel alignment of nearby cells can clearly be observed. Thus, single-cell segmentation provided precise information on how cells in dense colonies are distributed.

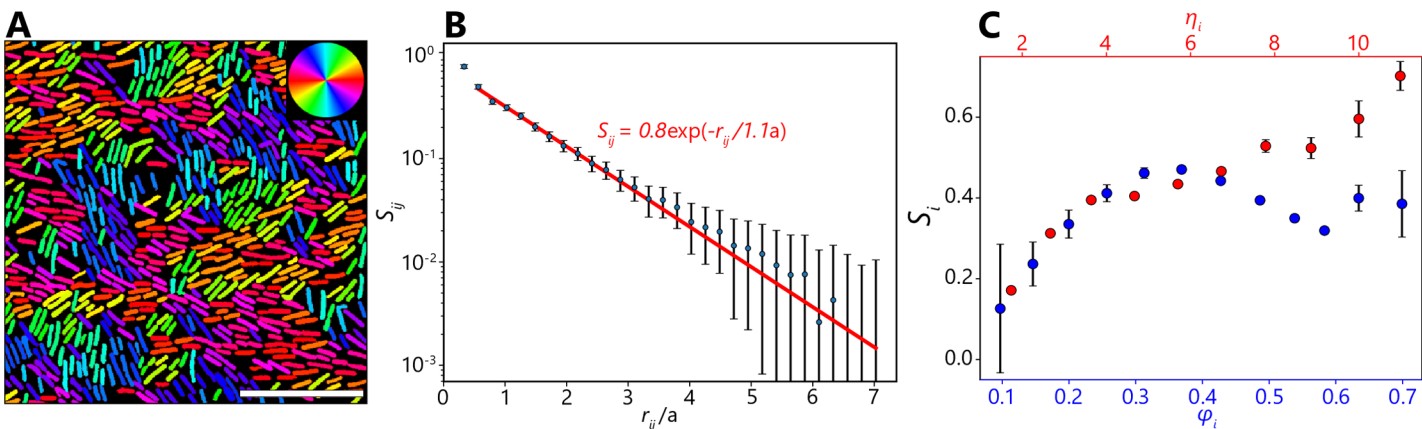

**Fig 4. Orientational order in dense *Pseudomonas* monolayers.** (A) Representative region of densely packed *P. a.*, segmented using a model trained on processed synthetic images and colored according to their angles relative to the vertical axis, showing parallel alignment of nearby cells. Scale bar is 10 μm. (B) Alignment of pairs of cells ($N > 10^7$), measured by the order parameter $S_{ij}$, as a function of their dimensionless separation, $r_{ij}/a$, where $a$ is the mean cell length. Exponential fitting curve is shown in red. Error bars represent variation between different cell pairs at each distance. (C) Local cell alignment ($N = 51262$) as a function of dimensionless packing fraction $\phi_i$ (blue points and bottom x-axis) and cell aspect ratio $\eta_i$ (red points and top x-axis). The packing fraction for each cell is the proportion of non-zero pixels in the segmentation mask within a radius of $2a$ around the cell, where $a$ is the mean cell length. The local alignment for the $i^{th}$ cell, $S_i$, is the average of the order parameter $S_{ij}$ for all cell pairs $i, j$ that include the $i^{th}$ cell. The data analyzed here are extracted from images taken in 2 separate experiments performed using the same protocol.

It was found that local cell alignment decays exponentially with a correlation length comparable to the length of a single cell. For any pair of cells separated by a distance $r_{ij} = \sqrt{(x_i - x_j)^2 + (y_i - y_j)^2}$, where $(x_i, y_i)$ is the position of the $i^{th}$ cell, the alignment is quantified via the order parameter

$$S_{ij} = 2\cos^2(\theta_i - \theta_j) - 1, \tag{2}$$

where $\theta_i$ is the angle of the $i^{th}$ cell measured relative to the vertical axis. Pairs of cells parallel/perpendicular to each other have $S_{ij} = \pm 1$, respectively. The dimensionless cell separation is determined by dividing $r_{ij}$ by the mean cell length $a = 2.8$ μm. Calculating $r_{ij}/a$ and $S_{ij}$ for every pair of cells with $r_{ij} < 20$ μm in our images ($N > 10^7$) shows how the orientational order of the cells depends on distance (Fig 4B). We find that $S_{ij}$ decays exponentially as a function of $r_{ij}/a$. Fitting gives a correlation length of $1.1a$ or $3.2$ μm (slightly shorter than the length of the average *P. a* cell).

Local cell alignment depends non-monotonically on local cell density. For each cell analyzed ($N = 51262$), the nematic order parameter,

$$S_i = \sum_j S_{ij} = \sum_j 2\cos^2(\theta_i - \theta_j) - 1, \tag{3}$$

is calculated to measure the average local alignment of each cell to its neighbors (defined as all cells within one correlation length, $3.2$ μm). Additionally, the dimensionless local cell density, or packing fraction $\varphi_i$, is calculated for each cell. The packing fraction represents the fraction of the PDMS surface covered by bacteria, and is determined by measuring the proportion of non-zero pixels in the segmentation mask within a radius of $2a$ around the cell, where $a = 2.87$ μm is the mean cell length. The packing fraction varied from 0.05 to 0.74, and the local cell alignment varied from −1.0 to 1.0. For low packing fractions ($\varphi \lesssim 0.4$), cell alignment increases with increasing cell density. At higher densities, single cell alignment remains

approximately constant, on average, and even decreases slightly on the interval $0.4 \lesssim \varphi \lesssim 0.6$ (Fig 4C, blue).

Cell alignment increases with increasing cell elongation. The shape of each cell was quantified by measuring the aspect ratio $\eta_i$ (cell length divided by width) of each segmented cell. Aspect ratios varied from 1.01 to 13.7, although there were not enough cells with $\eta_i > 11$ to calculate reliable statistics. The nematic order parameter $S_i$ increased monotonically with $\eta_i$ (Fig 4C, red). Results such as these require individual cell dimensions, positions, and orientations to be measured simultaneously in dense colonies, which is only possible here thanks to the accurate segmentation provided by our model trained on processed synthetic images.

### Simultaneous segmentation and classification of multi-species colonies

Segmentation models trained on processed synthetic images can achieve single-cell segmentation and classification of multi-species colonies. Suspensions of *P. a.* and *S. a.* are mixed and stained with SYTO 9 as described above, and imaged using a confocal laser scanning microscope. Two separate segmentation models are trained in Omnipose on synthetic images processed by a cycleGAN, and combined in a single Python script to identify both species (Fig 5A). On average, the model accurately identifies and segments 85% and 84% of cells for *S. a.* and *P. a.*, respectively (Fig 5B). Neither model systematically under- or oversegments cells, on average.

Multi-species segmentation is also possible for unstained mixed colonies imaged using brightfield microscopy. Mixed suspensions are simply deposited on a coverslip and imaged with no staining or special sample preparation (Fig 5C). A separate set of segmentation models is trained for this case from the ones used for stained bacteria. For *S. a.* and *P. a.*, the model accurately identifies 89% and 87% of cells, on average, respectively, which is slightly better than the segmentation performed on confocal images (Fig 5D). Neither of these models systematically over- or undersegments cells. These results demonstrate both the power and adaptability of segmentation models trained on processed synthetic images processed by cycleGAN, as these can easily be trained for a variety of imaging techniques and sample preparation methods.

## Discussion

### Key advantages of this method

Using synthetic microscopy images processed with cycleGANs, we show that single-cell image segmentation models can be efficiently created for a variety of experimental setups and imaging methods without tedious manual annotation. Accurate cell segmentation and classification is achieved even when cells are densely packed (that is, cells are touching or overlapping), when multiple species of different shapes are mixed, and when bacteria are grown on substrates that are not optimized for high-resolution imaging.

This approach has several advantages over other ways to create segmentation models. First, it does not require any human annotation of training images, which is not only very time-consuming but can also introduce human biases into the training data [40,41]. Second, the use of synthetic images allows prior knowledge of the system to be integrated into the segmentation model. Here, models are trained to detect cells with specific morphologies in images where cell boundaries are ambiguous as a result of noisy images. As previous studies have shown, the precise morphology and size of cell masks in the training data is extremely important to make sure cell dimensions can be accurately measured after segmentation [40].

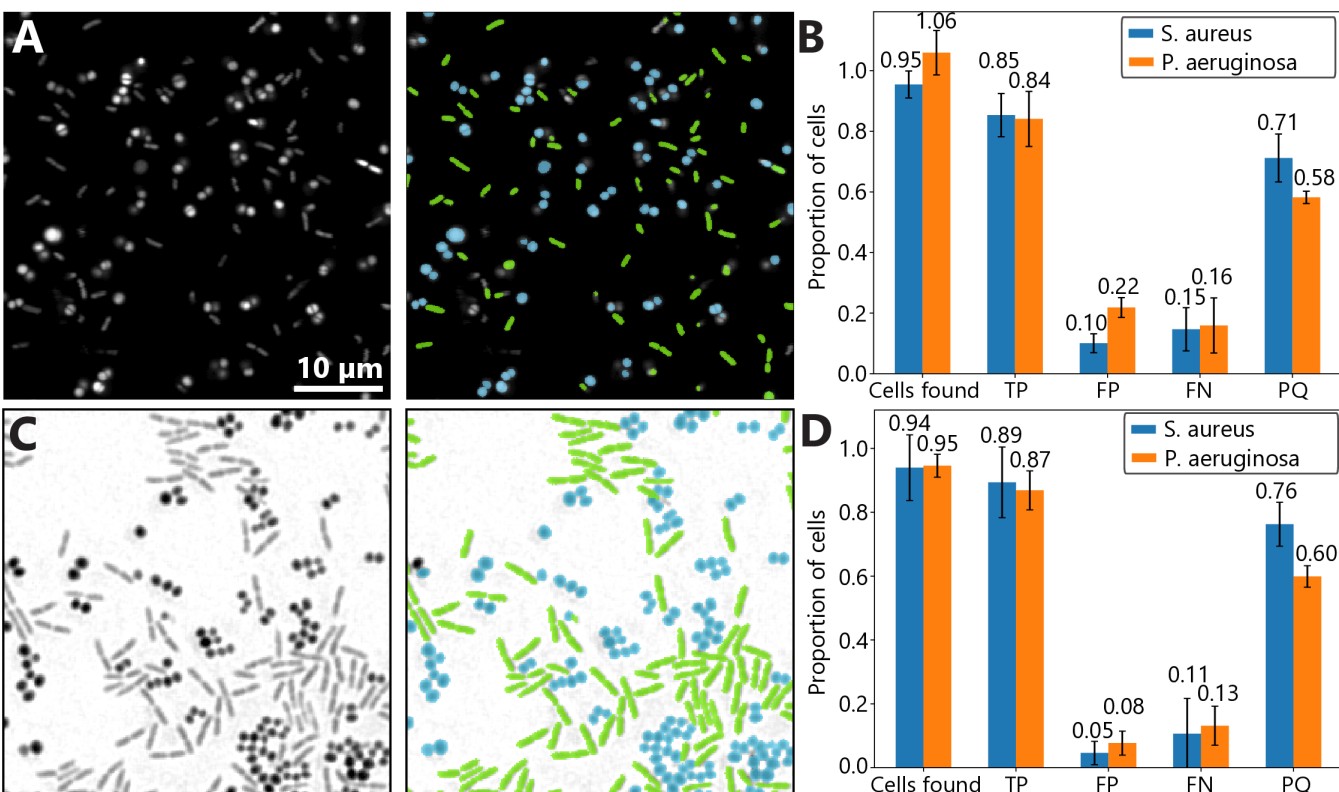

**Fig 5. Simultaneous segmentation and classification of *P. aeruginosa* and *S. aureus*.** (A) Confocal microscopy image (63×) of a mixed bacterial suspension stained with SYTO 9. In the right panel, green and blue cells represent individual bacteria classified as *P. a.* and *S. a.*, respectively, by a model trained on synthetic microscopy images. The scale bar is 10 μm, and the scale is the same for all images. (B) Segmentation statistics for multi-species segmentation of confocal microscopy images. True positives (TP), false positives (FP), and false negatives (FN) are given as a proportion of the number of cells in the ground truth mask, and panoptic quality (PQ) [38] is defined between 0 and 1. Error bars represent standard error from variation between images. The data are drawn from 4 different images, such as the ones in panel (A), and include 510 cells in total. (C) Brightfield microscopy image (40×) of a mixed bacterial suspension. In the right panel, green and blue cells represent individual bacteria classified as *P. a.* and *S. a.*, respectively, by a model trained on synthetic microscopy images. (D) Segmentation statistics for multi-species segmentation of images from brightfield microscopy. Statistics are given as a proportion of the number of cells in the ground truth mask. Error bars represent standard error from variation between images. The data are drawn from 6 different images, such as the ones in panel (C), and include 233 cells in total.

Third, the use of image-to-image translation via cycleGANs greatly simplifies the creation of realistic synthetic images suitable for the training of segmentation models. The only knowledge required for this step is the approximate geometry and concentration of cells in the images; no explicit modeling of the biology, physics, or optics of the experiment is required. These features make creating highly specialized segmentation models for different experimental setups highly efficient. The results described here demonstrate that the method is effective for both confocal laser scanning and brightfield microscopy, underscoring its versatility. Previously, synthetic images created with cycleGAN have also been used to train image analysis models for scanning transmission electron microscopy [18]. Together, these studies show that the approach can be effective for a variety of imaging modalities. In general, this adaptable method could be used by researchers from a wide variety of backgrounds for image analysis applications in the life sciences and beyond.

## Current limitations and potential extensions

Further improvements to synthetic images could lead to higher segmentation accuracy. The importance of cell shape is underscored by our results on multi-species segmentation in confocal images. While *S. a.* cells were modeled as perfect circles, they often appeared more oval-shaped or amorphous in real images (particularly during cell division), leading to a higher rate of false negatives from the model trained to identify circular cells and false positives from the model trained to identify non-circular cells (since the latter model would erroneously detect *S. a.* cells with higher aspect ratios). Future models will address this limitation by giving *S. a.* more diverse shapes in the raw synthetic images. Thanks to the high-quality image-to-image translation provided by cycleGANs, such changes to the synthetic images are relatively easy to implement, meaning the slightly higher false positive rate associated with the *P. a.* model for confocal microscopy images of mixed colonies is not a fundamental limitation of our approach.

The use of GANs causes processed synthetic images to occasionally contain artifacts such as additional, faint cells that are absent in the original masks, as seen in Fig 1C. Ensuring that the cell concentration in the raw synthetic images is high already minimizes this problem, as demonstrated by the low false negative rate, but further improvements to cycleGAN training may be necessary to remove the hallucinated cells altogether.

We expect this methodology to facilitate the segmentation of more species with complex shapes, which has been a key challenge in the study of the spatiotemporal development of biofilms [15]. However, the approach presented here cannot distinguish different bacteria strains with identical morphologies (e.g. mutant strains of a single species), and further testing will need to be carried out to determine how accurately bacteria of similar morphologies (e.g. two rod-shaped species of different sizes) can be classified. Various cell modeling methods already exist in the literature [13,21–23,42,43], which may be used to create raw synthetic images containing cells with complex and diverse morphologies.

Slight differences between synthetic images and real images will always remain, potentially limiting the accuracy of segmentation models trained on synthetic data alone. Therefore, it would be interesting to train models on a combination of hand-annotated real images and synthetic images to determine whether doing so can further improve their accuracy. Transfer learning could be an especially interesting approach, where a model is pre-trained on large synthetic datasets first and refined using real, hand-annotated images. The pre-trained model could be used to greatly speed up annotation by producing masks which only need to be corrected slightly by hand.

For all models developed here, the final quality of the segmentation can be further improved by introducing additional post-processing steps to the masks produced by the segmentation model. Cell masks can be filtered by size, shape, and position in a variety of ways depending on the experimental system or the desired information to be extracted. For example, principal component analysis can be used to categorize cell masks by their shape and isolate ones that were likely segmented incorrectly for further post-processing [21]. Additionally, if segmentation is combined with tracking, additional metrics become available to correct potential segmentation errors [12]. It should be noted that the segmentation framework used here, Omnipose, was not designed for the purpose of multi-species classification. It is possible that a different network architecture could provide better results with the training data produced here. In particular, an interesting alternative is to train a single segmentation model on synthetic images to segment all the cells in the images, and then separately train a classification model (for example using one-hot encoding) to identify the species of each cell. These post-processing steps and the development of more specialized tools for segmenting mixed

bacterial colonies lie beyond the scope of this work, whose primary purpose is to advance our understanding of bacterial self-organization and to show that synthetic microscopy images processed by cycleGANs provide an excellent replacement for hand-annotated training data that greatly increases the efficiency with which segmentation models are created.

Since the method presented here is rapid, versatile, and only requires the training of neural networks using widely available and well-established methods (cycleGAN and Omnipose, or other similar segmentation algorithms), the barrier to entry is minimal, making it an ideal choice even for researchers with minimal computational experience. This approach is highly synergistic with other recent advances in bacterial segmentation, which stand to benefit from the rapid creation of training data without complex modeling, as demonstrated here.

## Impact on the study of bacterial self-organization

The ability to accurately perform single-cell segmentation in different environments is important to understand the self-organization of bacteria and, in turn, shed light on the properties of dangerous biofilms. In particular, it enables the simultaneous determination of cell positions, orientations, and morphologies, as demonstrated here. Segmentation of dense *P. a.* monolayers revealed quantitative information about their self-organization. The correlated alignment between nearby cells was found to decay exponentially over distances up to 7 times the average cell length $a$, with a correlation length of $1.1a$. This relatively rapid spatial decay of cell alignment is qualitatively consistent with simulations of hard, rod-shaped bacteria of similar aspect ratios to those in our experiments [44]. At low packing fractions, cell alignment increases with cell density. Such an increase is expected from the Onsager theory of liquid crystals, and could represent a continuous phase transitions between an isotropic (disordered) phase and a nematic crystalline phase [45]. This density-dependent orientational ordering is consistent with simulations of bacterial colonies, where cell alignment is driven by steric interactions that become stronger when cells are densely packed [5,25]. Additionally, our results show a clear, monotonic dependence of alignment on cell elongation. While such behavior is expected in liquid crystals, to our knowledge, it has not previously been shown experimentally in bacterial monolayers.

Further work using single-cell segmentation of bacteria is needed to better understand the dependence of their nematic ordering on cell morphology and packing, particularly at high cell densities. Additionally, our approach would be ideal to determine how surface properties, cell motility, or growth medium composition affect this orientational ordering, since the use of synthetic images processed by cycleGAN enables us to rapidly train segmentation models for diverse imaging conditions. Another important extension of the work presented here is to develop similarly efficient segmentation models in 3D and investigate the effects of more complex geometries on bacterial colonies. Work on this problem is currently underway and will enable the analysis of bacterial aggregates in more varied and relevant settings.

In real infections, bacterial species rarely act alone, and the interactions between multiple species must be better understood in order to develop effective new therapies, particularly alternatives to antibiotics. Single-cell, multi-species segmentation is an important component of this challenge, as it is necessary for understanding mechanical interactions in realistic microbial communities. When designing experiments to investigate bacterial infections in clinically relevant systems, it is crucial to minimize the complexity of sample preparation methods and to avoid the addition of multiple fluorescent stains during bacterial growth or the use of fluorescent mutant strains, since these are not present in vivo. Thus, the ability to distinguish multiple species that are either unstained (using brightfield microscopy) or stained with a single fluorescent marker is highly useful. Species classification tools have

already been developed to identify bacterial strains in images with great accuracy for high-throughput identification of pathogens [46,47]. However, these existing methods only work when the images contain a single species. We are convinced that the methods developed here will pave the way towards more versatile tools that can be applied to a wide range of clinically relevant scenarios, including diagnostics and antibiotic susceptibility testing.

## Supporting information

**S1 Appendix. Supplemental methods and analysis.** Detailed description of synthetic image generation and training procedure for segmentation models and cycleGAN. Also contains analysis of how accurately cell properties are measured using different segmentation models as a function of IoU.
(PDF)

## Acknowledgments

We would like to thank Dr. Qun Ren for fruitful discussions and scientific input, and her team for their support in the lab. We also thank Georgeos Hardo for kindly providing the segmentation model trained on synthetic images from SyMBac.

## Author contributions

**Conceptualization:** Vincent Hickl, Abid Khan, René M. Rossi, Bruno F. B. Silva, Katharina Maniura-Weber.

**Formal analysis:** Vincent Hickl, Abid Khan.

**Funding acquisition:** René M. Rossi, Katharina Maniura-Weber.

**Investigation:** Vincent Hickl, Abid Khan.

**Methodology:** Vincent Hickl, Abid Khan, Bruno F. B. Silva.

**Software:** Vincent Hickl, Abid Khan.

**Supervision:** René M. Rossi, Bruno F. B. Silva, Katharina Maniura-Weber.

**Visualization:** Vincent Hickl, Abid Khan, Bruno F. B. Silva.

**Writing – original draft:** Vincent Hickl, Abid Khan.

**Writing – review & editing:** Vincent Hickl, Abid Khan, René M. Rossi, Bruno F. B. Silva, Katharina Maniura-Weber.

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
