## [Decision Letter · Decision Letter 0]

5 Dec 2024

PCOMPBIOL-D-24-01793

Segmentation of dense and multi-species bacterial colonies using models trained on synthetic microscopy images

PLOS Computational Biology

Dear Dr. Hickl,

Thank you for submitting your manuscript to PLOS Computational Biology. After careful consideration, we feel that it has merit but does not fully meet PLOS Computational Biology's publication criteria as it currently stands. Therefore, we invite you to submit a revised version of the manuscript that addresses the points raised during the review process.

Please submit your revised manuscript within 60 days Feb 04 2025 11:59PM. If you will need more time than this to complete your revisions, please reply to this message or contact the journal office at ploscompbiol@plos.org. Please include the following items when submitting your revised manuscript:

We look forward to receiving your revised manuscript.

Kind regards,

Virginie Uhlmann

Academic Editor

PLOS Computational Biology

Amber Smith

Section Editor

PLOS Computational Biology

Feilim Mac Gabhann

Editor-in-Chief

PLOS Computational Biology

Jason Papin

Editor-in-Chief

PLOS Computational Biology

**Journal Requirements:**

**Reviewers' comments:**

Reviewer's Responses to Questions

**Comments to the Authors:**

**Please note that the review is uploaded as an attachment.**

Reviewer #1: • 23-24: Great summary of the problem

• 45-46: segmentation alone would never be expected to distinguish between species, only a separate classification network would do this. But yes, we absolutely want to train models that can segment any set of species simultaneously regardless of cell-to-cell variation in signal or morphology.

• 64: very exciting promise!

• 71: what does this mean? Back-diluted and grown to 0.2?

• 145: a great point, and it is good that it is made both for GAN and U-net

• 167-175: this is an unexpected approach. I would have thought that training to exclude some cells and include others would cause the model to not learn very well. It makes sense that a number of cells would be false positives, because Omnipose does some stretching of the images and crop reflections to simulate varied morphologies. I think that a separate model trained just for one-hot encoding of species used in conjunction with a global segmentation model would yield fewer false positives.

• 180: PQ is very similar to the Jaccard index. This should probably be mentioned. It still has a maximum of 1, but the false negatives and false positives have half the effect as IOU cutoffs increase. Would love to see a PQ vs IoU cutoff curve.

• 192: a very clear and reasonable set of parameters for the image analysis,

• 261: I wonder how important modeling cell orientation at a global scale is. No doubt, we should have all kinds of interfaces represented in the dataset, but I would love to see models trained with and without global structure to see if there is anything gained by making this constraint in the image. My hunch is that it is more important to model ambiguous cell interfaces that occur from division.

• 262-270: how do we know that the simulated cells match the cell boundaries? I think that a test against the symbac fluorescence model is critical (https://doi.org/10.1038/s44303-024-00024-4), as that was specifically designed to be physiologically accurate.

• 307: IoU of 0.5 is frankly a meaningless cutoff. Only 0.8 or higher should be used for true positives. Fig 3E shows that most of the IoUs are >0.5, so this should not affect the qualitative results of the paper, but it is an important quantitative distinction.

• 309: the most recent state of the art model is actually the aforementioned symbac model (granted, this was only published in September). This should be shown in addition to or instead of bact_fluor_omni, which was shown to dilate cell masks (this was an artifact of elastic local registration in the training set).

• 462: I think that the tactic of cell simulation will indeed be key to developing cell classification based on signal and morphology. However, training separate segmentation models is likely not the best option. Training one-hot encoding models will be more efficient and result in fewer false positives, and this technique of simulation is absolutely critical to training models to do so (as the number of classes with change with the number of prediction species). The possibility that alternative models would be better for this task is discussed in 470-471.

Fig 1:

• There are a LOT of artifacts in the simulated images. Linear artifact probably a result of tiling and lack of blending. But there are other artifacts in the PSF.

• I suggest using the ncolor package to use shades of green for Pa and shades of red for Sa.

FIg 2:

• The IoU panel only shows one type of offset that would cause less than 1. I suggest showing other kinds of errors, as I doubt the most relevant error here for bact_fluor_omni is the offset. I think it would be a dilation.

FIg 3:

• I would maybe use one shade for masks that stayed the same and red etc. for masks that changes substantially, so that it is clear what was correct. Or maybe a difference mask.

Data: there is an oversight in the data availability. The masks and binarized images are in the synthetic images folder on zenodo, but not the actual cycleGAN output.

Reviewer #2: Review uploaded as attachment.

Reviewer #3: This article proposes a strategy for training microscopy image segmentation models from transformed synthetic binary masks. This manuscript could have been potentially interesting but I have several major comments:

1 - In my opinion, this work does not demonstrate that synthetic image production is strictly necessary to improve CNN-type model training. Indeed, there are no results showing that a simple modification of synthetic masks (such as Gaussian filter and noise) is not sufficient to train a CNN model and thus provide better results than pre-trained models.

2 - This work does not demonstrate that training with the GAN-transformed synthetic masks (circles and rods) improves results obtained for different morphologies (crescents, chains, curved rods, etc.). Moreover, if the user must generate their own synthetic binary masks for the morphologies of interest, no solution is proposed in this direction. These two essential aspects for applying this method to real cases of morphological diversity are not discussed. And precisely, the authors insist on the need for universal solutions for bacterial mixtures in real community study conditions. The work presented in this article absolutely requires results obtained with real bacterial images presenting a true diversity of shapes (Caulobacter, Anabaena, Streptococcus, Spirochaetota, morphological mutants etc.).

3 - The authors base their comparison on the improvement in results compared with a single example of a pre-trained model, namely “Bact_fluor_omni”. This comparison is not sufficient for two reasons: firstly, only a pre-trained model for segmentation from fluorescence images was tested, but many light microscopy explorations use phase contrast images as the source for cell segmentation (the omnipose package provides a pre-trained model for phase contrast). The GAN process of synthetic image should be able to mimic the microscopy phase contrast image in order to train a model with. Secondly, the only type of pre-trained model tested is a CNN-type model, whereas currently models based on “transformers” seem to have a better ability to adapt to the diversity of objects in the image (this is illustrated in these articles which are not mentioned in this work: Ma et. al Nat Methods . 2024 June ; 21(6): 1103–1113. doi:10.1038/s41592-024-02233-6, Gihun et al. Mediar: Harmony of data-centric

and model-centric for multi-modality microscopy. In Proceedings of The Cell Segmentation

Challenge in Multi-modality High-Resolution Microscopy Images, volume 212, pages 1–16,

2023).

**Have the authors made all data and (if applicable) computational code underlying the findings in their manuscript fully available?**

Reviewer #1: Yes

Reviewer #2: None

Reviewer #3: Yes

PLOS authors have the option to publish the peer review history of their article (what does this mean?). If published, this will include your full peer review and any attached files.

Reviewer #1: **Yes: **Kevin John Cutler

Reviewer #2: **Yes: **Teresa W. Lo

Reviewer #3: No

**Figure resubmission:**
---

## [Decision Letter · Decision Letter 1]

28 Jan 2025

PCOMPBIOL-D-24-01793R1

Segmentation of dense and multi-species bacterial colonies using models trained on synthetic microscopy images

PLOS Computational Biology

Dear Dr. Hickl,

Thank you for submitting your manuscript to PLOS Computational Biology. After careful consideration, we feel that it has merit but does not fully meet PLOS Computational Biology's publication criteria as it currently stands. Therefore, we invite you to submit a revised version of the manuscript that addresses the points raised during the review process.

Please submit your revised manuscript within 30 days Mar 30 2025 11:59PM. If you will need more time than this to complete your revisions, please reply to this message or contact the journal office at ploscompbiol@plos.org. Please include the following items when submitting your revised manuscript:

We look forward to receiving your revised manuscript.

Kind regards,

Amber M Smith

Section Editor

PLOS Computational Biology

Amber Smith

Section Editor

PLOS Computational Biology

**Additional Editor Comments :**

The reviewers noted significant improvements in the last revision. There are a few items they would still like to be addressed. First, Reviewer #1 has remaining concerns about the code and image availability, in addition to minor clarifications. Second, Reviewer #2 felt their initial criticisms were not adequately addressed and have suggested investigation into diversified morphologies with possible testing of the results in phase contrast. Their argument is that phase contrast is a widely used and that and adding phase contrast tests would require minimal experimental effort. If it is possible and agreed that it would improve the manuscript, please perform as suggested. If it is not possible or you disagree with the reviewer, please articulate that in your response and add appropriate caveats to the discussion.

**Reviewers' comments:**

Reviewer's Responses to Questions

Reviewer #1: The responses to all reviewers was extremely thorough and made for substantial improvements to the manuscript.

A last weakness of this manuscript worth mentioning is that we do not have a head-to-head comparison of a segmentation model trained on cycleGAN images versus one trained on corresponding real images. The authors are careful to mention that the bact_phase_omni and SyMBac models are trained on distinct datasets not representative of the test dataset. We should of course expect that those models would perform worse than any model trained using real, representative data. The huge innovation in this work is that no manual annotation was required to train a new segmentation model on such a distinct dataset. However, the fact that it performs better than the other models is not particularly meaningful - a segmentation model trained on apples should not be expected to segment oranges.   Instead of training cycleGAN on their own data, the authors could instead train on the same training datasets of the aforementioned models. This would eliminate the variable of train vs test data representation and show that it is possible to reach X% of the segmentation accuracy without manual annotation. [There is the subtlety that this experiment uses manual masks instead of generated masks (“synthetic images” in the manuscript), but generation of arbitrary cell morphologies is outside the scope of the paper.]  The principal claim of this work is that the cycleGAN produces good enough training data for a segmentation model to be useful. How useful depends on the scientific question to be answered, and the authors demonstrate their use case well. Other use cases would require far higher segmentation accuracy, and this work does not give such a limit. Although the manuscript feels incomplete without this comparison, the advancement it demonstrates is sufficient for publication in my opinion.   For that advancement to be useful to anyone, documentation must be written to guide users through training a cycleGAN on their own data. Hardware requirements, package versions, training image folder structure and file naming, etc. must all be specified in the GitHub README. A commented .py script or a jupyter notebook should be provided to run the training. Expected training times would also be helpful.

As a last minor comment: “synthetic image” vs “processed synthetic image” is still confusing. Consider using “binary mask” or “generated masks” for the procedurally generated, binarized cell labels and reserve “synthetic image” for the output of the cycleGAN. This would be more consistent with existing literature.

Reviewer #2: Excellent work! This is very exciting and a helpful contribution to the field.

Reviewer #3: This manuscript is of real interest in the field of bacterial microscopy image segmentation.

The authors have improved the manuscript taking into account the reviewers' remarks. The revised version highlights the strengths and limitations of this work.

But on the substance of the paper, it could have had a greater impact if the authors had diversified the morphologies of the synthetic images and if they had carried out phase contrast tests. Indeed, there is no evidence to suggest that the method described here improves results for morphologies other than rod morphologies. The authors consider that tests with fluorescence images (confocal and wide-field) validate their method for all microscopy modalities. I believe that tests with phase contrast images and the omnipose phase contrast model would have greatly increased the scope of this work and its interest for microbiologists who make extensive use of phase contrast.

**Have the authors made all data and (if applicable) computational code underlying the findings in their manuscript fully available?**

Reviewer #1: **No: **Zenodo/github needs to be updated to include the 13 images and labels used for PQ evaluation. As mentioned, github repos need documentation for use of the code.

Reviewer #2: Yes

Reviewer #3: Yes

PLOS authors have the option to publish the peer review history of their article (what does this mean?). If published, this will include your full peer review and any attached files.

Reviewer #1: **Yes: **Kevin John Cutler

Reviewer #2: **Yes: **Teresa W. Lo

Reviewer #3: No

**Figure resubmission:**
---

## [Editor Report · Decision Letter 2]

13 Feb 2025

Dear Dr. Hickl,

We are pleased to inform you that your manuscript 'Segmentation of dense and multi-species bacterial colonies using models trained on synthetic microscopy images' has been provisionally accepted for publication in PLOS Computational Biology.

Best regards,

Amber M Smith

Section Editor

PLOS Computational Biology

---

## [Editor Report · Acceptance letter]

PCOMPBIOL-D-24-01793R2

Segmentation of dense and multi-species bacterial colonies using models trained on synthetic microscopy images

Dear Dr Hickl,

I am pleased to inform you that your manuscript has been formally accepted for publication in PLOS Computational Biology. Your manuscript is now with our production department and you will be notified of the publication date in due course.

With kind regards,

Lilla Horvath
